# DISTILLATION-FREE ONE-STEP DIFFUSION FOR REAL-WORLD IMAGE SUPER-RESOLUTION

## ABSTRACT

Diffusion models have been achieving excellent performance for real-world image super-resolution (Real-ISR) with considerable computational costs. Current approaches are trying to derive one-step diffusion models from multi-step counterparts through knowledge distillation. However, these methods incur substantial training costs and may constrain the performance of the student model by the teacher's limitations. To tackle these issues, we propose DFOSD, a Distillation-Free One-Step Diffusion model. Specifically, we propose a noise-aware discriminator (NAD) to participate in adversarial training, further enhancing the authenticity of the generated content. Additionally, we improve the perceptual loss with edge-aware DISTS (EA-DISTS) to enhance the model's ability to generate fine details. Our experiments demonstrate that, compared with previous diffusion-based methods requiring dozens or even hundreds of steps, our DFOSD attains comparable or even superior results in both quantitative metrics and qualitative evaluations. Our DFOSD also attains higher performance and efficiency compared with other one-step diffusion methods. We will release code and models.

## 1 INTRODUCTION

Real-world image super-resolution (Real-ISR) is a challenging task that aims to reconstruct high-resolution (HR) images from their low-resolution (LR) counterparts in real-world settings (Wang et al., 2020). Most image super-resolution (SR) methods (Kim et al., 2016; Johnson et al., 2016; Ledig et al., 2017; Chen et al., 2022; 2023) use Bicubic downsampling of HR images to generate LR samples for training and testing models. These methods achieve good results in reconstructing simple degraded images. However, they struggle with the complex and unknown degradations widely existing in real-world scenarios. Moreover, these methods often amplify the noise in LR images during reconstruction. Previous research has predominantly employed generative adversarial networks (GANs) (Goodfellow et al., 2020) architectures for image SR tasks (Wang et al., 2021a; Zhang et al., 2021; Liang et al., 2021). However, these approaches often struggle to train models that accurately capture real-world data distributions, leading to suboptimal generated content. Diffusion models (DMs), known for their strong denoising capabilities and ability to model data distributions, have been widely adopted in the field of image generation in recent years. Recently, numerous super-resolution (SR) methods based on pre-trained diffusion models have exhibited outstanding performance by leveraging their powerful priors and generative capabilities.

Specifically, recent real-world image super-resolution (Real-ISR) models have predominantly leveraged powerful pre-trained diffusion models, such as large-scale text-to-image (T2I) models like Stable Diffusion (Wu et al., 2024b; Yang et al., 2024; Lin et al., 2024). With training on billions of image-text pairs and a capacity to model complex data distributions, these pre-trained T2I models provide extensive priors and powerful generative abilities. Most diffusion model (DM)-based methods generate high-resolution (HR) images by employing ControlNet models (Zhang et al., 2023), conditioning on the low-resolution (LR) inputs. However, these methods typically require tens to hundreds of diffusion steps to produce high-quality HR images. The introduction of ControlNet not only increases the number of model parameters but also further exacerbates inference latency. Consequently, DM-based multi-step diffusion methods often incur delays of tens of seconds when processing a single image, which significantly limits their practical application in real-world scenarios for low-level image reconstruction tasks, such as Real-ISR.

To accelerate the generation process of diffusion models, recent research has introduced numerous one-step diffusion methods, known as diffusion distillation, which distill multi-step pre-trained diffusion

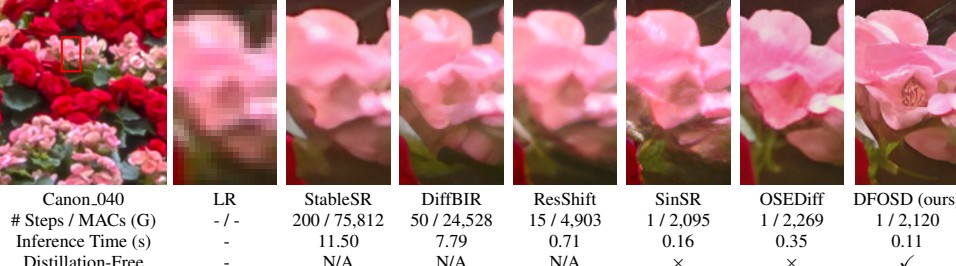

| | Canon_040 | LR | StableSR | DiffBIR | ResShift | SinSR | OSEDiff | DFOSD (ours) |
|---|---|---|---|---|---|---|---|---|
| # Steps / MACs (G) | | - / - | 200 / 75,812 | 50 / 24,528 | 15 / 4,903 | 1 / 2,095 | 1 / 2,269 | 1 / 2,120 |
| Inference Time (s) | | - | 11.50 | 7.79 | 0.71 | 0.16 | 0.35 | 0.11 |
| Distillation-Free | | - | N/A | N/A | N/A | × | × | ✓ |

Figure 1: Visual comparisons (×4) of different DM-based Real-ISR methods, including their inference times and MACs (Multiply-Accumulate Operations), for an output size of 512×512. The inference times are measured on an A100 GPU. StableSR (Wang et al., 2024a), DiffBIR (Lin et al., 2024), and ResShift (Yue et al., 2024) are multi-step DM-based methods, performing 200, 50, and 15 sampling steps respectively. Our DFOSD is distillation-free when compared with other one-step diffusion models, like SinSR (Wang et al., 2024b) and OSEDiff (Wu et al., 2024a). Our DFOSD generates realistic details and achieves the lowest inference latency and MACs.

models into one-step counterparts. Most of these approaches employ a knowledge distillation strategy, using the multi-step diffusion model as a teacher to train a one-step diffusion student model. These methods significantly reduce inference latency, and the quality of the generated images can be comparable to that of multi-step diffusion models. Real-ISR methods based on one-step diffusion models have become an increasingly popular research direction, with representative methods such as SinSR (Wang et al., 2024b) and OSEDiff (Wu et al., 2024a). While these methods achieve promising visual results, the inclusion of the teacher network increases training overhead. The performance of the student network is often constrained by the teacher network.

To overcome the aforementioned challenges, we propose DFOSD, a novel approach that generates HR images from their corresponding LR inputs in a single sampling step. Unlike previous one-step diffusion SR models, we do not employ knowledge distillation to train our one-step diffusion generator. Our approach eliminates the need to leverage outputs or corresponding noise from multi-step diffusion models, allowing us to train solely on real-world datasets. This significantly reduces training overhead and overcomes the limitations imposed by teacher models. Furthermore, we do not utilize models like CLIP (Radford et al., 2021) to encode prompts as conditional inputs for the diffusion model. Instead, we train a learnable text embedding. This further reduces the model's inference time without compromising performance. As shown in Fig. 1, DFOSD not only achieves the best visual results but also attains the fastest inference speed.

To better leverage the prior knowledge of pre-trained multi-step models and enhance the authenticity of the generated images, we propose a noise-aware discriminator (NAD) initialized with parameters from the pre-trained stable diffusion (SD) UNet, which is trained adversarially alongside the generator. Specifically, our NAD takes the forward diffusion results of the latent features at various time steps, ensuring that its performance remains robust across different noise levels. NAD capitalizes on the prior knowledge of the pre-trained diffusion model, enhancing the reconstruction quality (see Fig. 1). Additionally, we propose edge-aware DISTS (EA-DISTS) loss to improve the authenticity of fine details in the generated content. Our comprehensive experiments indicate that DFOSD achieves superior performance and less inference time among one-step diffusion model (DM)-based Real-ISR models. When compared with multi-step DM-based models, DFOSD obtains comparable or even better performance with over 7× speedup in inference time (see Fig. 1).

Our main contributions are summarized as follows:

- We propose DFOSD, a Distillation-Free One-Step Diffusion SR model training paradigm. Our DFOSD significantly enhances the details and visual quality of generated images, achieving remarkable results in both evaluation metrics and visual assessments.

- We propose a noise-aware discriminator (NAD), which capitalizes on the prior knowledge from the pre-trained SD UNet and engages in adversarial training with the generator. Our NAD effectively enhances the realism and details of the reconstructed images.

- We improve the perceptual loss used in image SR model training by proposing the edge-aware DISTS (EA-DISTS) loss. Our EA-DISTS leverages image edges to enhance the model's ability and improve the authenticity of reconstructed details.

## 2 RELATED WORKS

### 2.1 REAL-WORLD IMAGE SUPER-RESOLUTION

Real-world image super-resolution (Real-ISR) aims to recover high-resolution (HR) images from low-resolution (LR) observations in real-world scenarios. The complex and unknown degradation patterns in such scenarios make Real-ISR a challenging problem (Ignatov et al., 2017; Liu et al., 2022a; Ji et al., 2020; Wei et al., 2020). To address this problem, models continuously evolve. Early image super-resolution models (Kim et al., 2016; Zhang et al., 2018c;b; Chen et al., 2022; 2023) typically rely on simple synthetic degradations like Bicubic downsampling for generating LR-HR pairs, resulting in subpar performance on real-world datasets. Later, GAN-based methods such as BSRGAN (Zhang et al., 2021), Real-ESRGAN (Wang et al., 2021a), and SwinIR-GAN (Liang et al., 2021) introduce more complex degradation processes. These methods achieve promising perceptual quality but encounter issues such as training instability. Additionally, they have limitations in preserving fine natural details. Recently, Stable Diffusion (SD) (Rombach et al., 2022b) is considered for addressing Real-ISR tasks due to its strong ability to capture complex data distributions and provide robust generative priors. Approaches such as StableSR (Wang et al., 2024a), DiffBIR (Lin et al., 2024), and SeeSR (Wu et al., 2024b) leverage pre-trained diffusion priors and ControlNet models (Zhang et al., 2023) to enhance HR image generation. While these methods significantly improve perceptual quality, the multi-step nature of diffusion models introduces latency issues, making them less practical for real-time applications in low-level image reconstruction tasks.

### 2.2 ACCELERATION OF DIFFUSION MODELS

Acceleration of diffusion models can reduce computational costs and inference time. Therefore, various strategies have been developed to enhance the efficiency of diffusion models in image generation tasks. Fast diffusion samplers (Song et al., 2021; Karras et al., 2022; Liu et al., 2022b; Lu et al., 2022a;b; Zhao et al., 2024) have significantly reduced the number of sampling steps from 1,000 to 15∼100 without requiring model retraining. However, further reducing the steps below 10 often leads to a performance drop. Under these circumstances, distillation techniques have made considerable progress in speeding up inference (Berthelot et al., 2023; Liu et al., 2022c; Meng et al., 2023; Salimans & Ho, 2022; Song et al., 2023; Zheng et al., 2023; Yin et al., 2024b; Liu et al., 2023; Geng et al., 2024). For instance, Progressive Distillation (PD) methods (Meng et al., 2023; Salimans & Ho, 2022) have distilled pre-trained diffusion models to under 10 steps. Consistency models (Song et al., 2023) have further reduced the steps to 2∼4 with promising results. Instaflow (Liu et al., 2023) further achieves one-step generation through reflow (Liu et al., 2022c) and distillation. Recent score distillation-based methods, such as Distribution Matching Distillation (DMD) (Yin et al., 2024c;a) and Variational Score Distillation (VSD) (Wang et al., 2024c; Nguyen & Tran, 2024), aim to achieve one-step text-to-image generation. They minimize the Kullback–Leibler (KL) divergence between the generated data distribution and the real data distribution. Although these approaches have made notable progress, they still face challenges, like high training costs and dependence on teacher models.

## 3 DISTILLATION-FREE ONE-STEP DIFFUSION (DFOSD)

In this section, we detail our Distillation-Free One-Step Diffusion (DFOSD) image super-resolution (SR) model. First, in Section 3.1, we review the fundamentals of diffusion models and introduce the principles underlying the DFOSD generator. Subsequently, we propose two key techniques for training our one-step diffusion SR model. In Section 3.2, we propose the noise-aware discriminator (NAD), which assesses image realism using the results of random forward diffusion applied to their latent representations. Then, in Section 3.3, we propose an improved perceptual loss function, edge-aware DISTS (EA-DISTS), designed to enhance the quality of image texture details. Finally, in Section 3.4, we outline the complete training process of the model.

### 3.1 PRELIMINARIES: DIFFUSION

Diffusion models include forward and reverse processes. During the forward diffusion process, Gaussian noise with variance $\beta_t \in (0,1)$ is gradually injected into the latent variable $z$: $z_t = \sqrt{\bar{\alpha}_t} z + \sqrt{1 - \bar{\alpha}_t} \epsilon$, where $\epsilon \sim \mathcal{N}(0, \mathbf{I})$, $\alpha_t = 1 - \beta_t$, and $\bar{\alpha}_t = \prod_{s=1}^{t} \alpha_s$. In the reverse process, we can directly predict the clean latent variable $\hat{z}_0$ from the model's predicted noise $\hat{\epsilon}$: $\hat{z}_0 = \frac{z_t - \sqrt{1 - \bar{\alpha}_t} \hat{\epsilon}}{\sqrt{\bar{\alpha}_t}}$, where $\hat{\epsilon}$ is the prediction of the network $\epsilon_\theta$ given $z_t$ and $t$: $\hat{\epsilon} = \epsilon_\theta(z_t; t)$.

As illustrated in Fig. 2, we first employ the encoder $E_\theta$ to map the low-resolution (LR) image $x_L$ into the latent space, yielding $z_L$: $z_L = E_\theta(x_L)$. Next, we perform a single denoising step to obtain

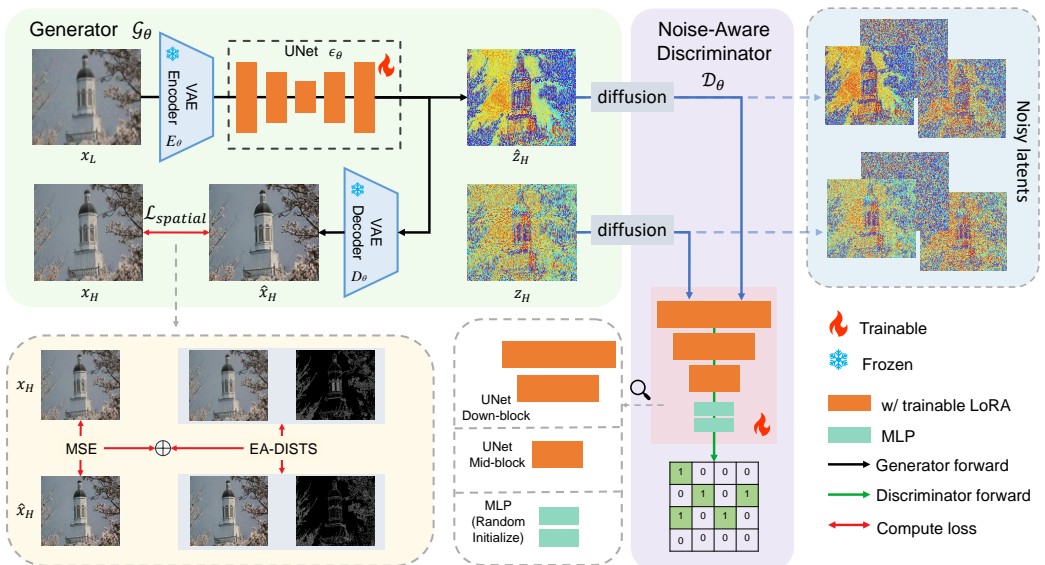

Figure 2: Training framework of DFOSD. The left side represents the generator $\mathcal{G}_\theta$, which includes the pre-trained VAE and UNet from Stable Diffusion. Only the UNet is fine-tuned using LoRA, while other parameters remain frozen. The right side depicts the noise-aware discriminator (NAD), which guides the training process without participating in inference. The NAD extracts the UNet Mid-block outputs and processes them through an MLP to generate realism votes for different image regions. Both the downsampling and middle blocks of the UNet in the discriminator are fine-tuned with LoRA, whereas the MLP is randomly initialized.

the predicted noise $\hat{\epsilon}$ and compute the high-resolution (HR) latent representation $\hat{z}_H$:

$$\hat{z}_H = \frac{z_L - \sqrt{1 - \bar{\alpha}_{T_L}}\,\epsilon_\theta(z_L; T_L)}{\sqrt{\bar{\alpha}_{T_L}}}, \tag{1}$$

where $\epsilon_\theta$ denotes the denoising network parameterized by $\theta$, and $T_L$ is the diffusion time step. Unlike one-step text-to-image (T2I) diffusion models (Song et al., 2023; Yin et al., 2024c), the input to the UNet of the Real-ISR diffusion models is not pure Gaussian noise. We set $T_L$ to an intermediate time step within the range $[0, T]$, where $T$ is the total number of diffusion time steps. In Stable Diffusion (SD), $T = 1,000$. Finally, we decode $\hat{z}_H$ using the decoder $D_\theta$ to reconstruct the HR image $\hat{x}_H$: $\hat{x}_H = D_\theta(\hat{z}_H)$. The entire computation process of the generator can be expressed as $\hat{x}_H = \mathcal{G}_\theta(x_L)$.

### 3.2 Noise-Aware Discriminator (NAD)

In an ideal scenario, we seek to achieve image restoration results that are almost indistinguishable from real images. Yet, training the generator directly without distillation often falls short of this goal. To improve the realism of generated images, we incorporate a discriminator. Training a discriminator from scratch, however, may result in unstable training dynamics, and converting the generator's latent outputs to pixel space for evaluation introduces considerable computational overhead. Stable Diffusion (SD), a robust pre-trained generative model with strong priors and a UNet-based architecture, provides a promising solution to these challenges. This inspires us to initialize the discriminator with pre-trained UNet parameters, perform operations directly in the latent space, and leverage the UNet bottleneck layer's robust information filtering and semantic condensation capabilities to construct the discriminator.

Figure 3 illustrates the visualization results of the latent representations of both generated and real images during the early stages of training. After undergoing forward diffusion at various random time steps, the UNet middle block outputs are visualized using dimensionality reduction techniques such as PCA, supervised UMAP, and LLE. The feature distributions of the generated images and real images exhibit distinct differences, thereby highlighting the Stable Diffusion (SD) UNet's robust information filtering and semantic condensation capabilities.

Based on these observations, we propose a noise-aware discriminator (NAD). To better leverage the diffusion model's ability to perceive noise at various levels and maintain the gap between generated and real images under different noise intensities, we feed the latent representations with randomly

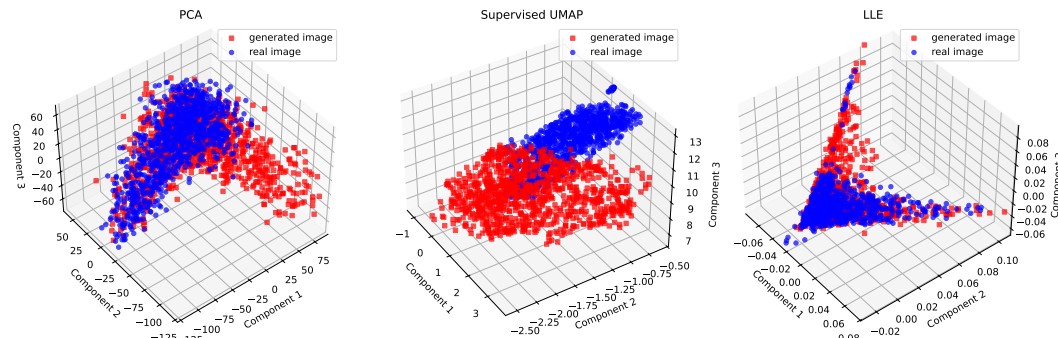

Figure 3: Visualization of features dimensionality reduction for the first 100 channels from the middle block outputs of the Stable Diffusion (SD) UNet. Notably, there is a significant difference in the feature distributions at the UNet's intermediate layers between real images and those generated by the one-step diffusion model during early training stages. This observation suggests that the intermediate layer features of the UNet are a robust basis for assessing image realism.

injected noise levels as inputs to the NAD. In Fig. 2, the NAD $\mathcal{D}_\theta$ consists of the UNet downsampling blocks (*i.e.*, UNet Down-block in Fig. 2) and middle block (*i.e.*, UNet Mid-block in Fig. 2), along with a MLP mapping the features into realism scores for different regions. $\mathcal{D}_\theta$ is initialized with the corresponding parameters of the SD UNet at the beginning of training. During training, we feed into the discriminator the forward diffusion results of both the latents predicted by the generator (*i.e.*, $\hat{z}_H$ as mentioned in Eq. 1) and the corresponding ground truth latent vectors $z_H = E_\theta(x_H)$.

The adversarial losses for updating the generator and discriminator are defined as:

$$\mathcal{L}_\mathcal{G} = -\mathbb{E}_{x_L \sim p_{\text{data}}, \, t \sim [0,T]} \left[ \log \mathcal{D}_\theta \left( F \left( \hat{z}_H, t \right) \right) \right], \tag{2}$$

$$\mathcal{L}_\mathcal{D} = -\mathbb{E}_{x_L \sim p_{\text{data}}, \, t \sim [0,T]} \left[ \log \left( 1 - \mathcal{D}_\theta \left( F \left( \hat{z}_H, t \right) \right) \right) \right]$$
$$- \mathbb{E}_{x_H \sim p_{\text{data}}, \, t \sim [0,T]} \left[ \log \mathcal{D}_\theta \left( F \left( z_H, t \right) \right) \right], \tag{3}$$

where $\hat{z}_H$ is computed as: $\hat{z}_H = \frac{z_L - \sqrt{1-\bar{\alpha}_T} \, \epsilon_\theta(z_L; T)}{\sqrt{\bar{\alpha}_T}}$, and $F(\cdot, t)$ denotes the forward diffusion process of $\cdot$ at time step $t \in [0, T]$, specifically,

$$F(z, t) = \sqrt{\bar{\alpha}_t} \, z + \sqrt{1 - \bar{\alpha}_t} \, \epsilon, \text{ with } \epsilon \sim \mathcal{N}(0, \mathbf{I}). \tag{4}$$

## 3.3 EDGE-AWARE DISTS

To further enhance the quality of the generated images, we aim to incorporate perceptual loss. Most image reconstruction methods utilize LPIPS (Learned Perceptual Image Patch Similarity) (Zhang et al., 2018a) as the perceptual loss. However, to better preserve image texture details and alleviate pseudo-textures in the reconstruction under higher noise levels, we need to focus on the textures on HR images. DISTS (Deep Image Structure and Texture Similarity) (Ding et al., 2020) can compute the structural and textural similarity of images, aligning with human subjective perception of image quality. Furthermore, regions with rich textures or details often exhibit strong edge information. Leveraging image edge information effectively enhances texture quality. Based on this, we propose a novel perceptual loss, termed Edge-Aware DISTS (EA-DISTS). This perceptual loss simultaneously evaluates the structure and texture similarity of the reconstructed and HR images and their edges, thereby enhancing texture detail restoration.

Our proposed EA-DISTS is defined as:

$$\mathcal{L}_{\text{EA-DISTS}}(\mathcal{G}_\theta(x_L), x_H) = \mathcal{L}_{\text{DISTS}}(\mathcal{G}_\theta(x_L), x_H) + \mathcal{L}_{\text{DISTS}}(\mathcal{S}(\mathcal{G}_\theta(x_L)), \mathcal{S}(x_H)), \tag{5}$$

where $\mathcal{S}(\cdot)$ represents the Sobel operator used to extract edge information from the images. It consists of two convolution kernels, $G_x$ and $G_y$, which detect horizontal and vertical edges, respectively:

$$G_x = \begin{bmatrix} -1 & 0 & 1 \\ -2 & 0 & 2 \\ -1 & 0 & 1 \end{bmatrix}, \quad G_y = \begin{bmatrix} -1 & -2 & -1 \\ 0 & 0 & 0 \\ 1 & 2 & 1 \end{bmatrix}. \tag{6}$$

The Sobel operator is applied to an image $x$ as follows:

$$\mathcal{S}(x) = \sqrt{(G_x * x)^2 + (G_y * x)^2}, \tag{7}$$

where $*$ denotes the convolution operation. This computation results in an edge map that highlights the structural and textural details of the image.

| Datasets | Metrics | Multi-step Diffusion | | | | One-step Diffusion | | |
|---|---|---|---|---|---|---|---|---|
| | | StableSR-s200 | DiffBIR-s50 | SeeSR-s50 | ResShift-s15 | SinSR-s1 | OSEDiff-s1 | DFOSD-s1 |
| RealSR | NIQE↓ | 4.8927 | 3.9472 | 4.5403 | 7.3495 | 5.7467 | 4.3443 | 3.9255 |
| | MUSIQ↑ | 60.53 | 68.02 | 66.37 | 56.18 | 61.62 | 67.31 | 69.21 |
| | ManIQA↑ | 0.5570 | 0.6309 | 0.6118 | 0.5004 | 0.5362 | 0.6148 | 0.6402 |
| | ClipIQA↑ | 0.5140 | 0.7295 | 0.6822 | 0.5848 | 0.6927 | 0.6827 | 0.6683 |
| RealSet65 | NIQE↓ | 4.9852 | 4.1218 | 4.6891 | 6.7303 | 5.6642 | 4.2245 | 3.9580 |
| | MUSIQ↑ | 58.89 | 71.23 | 69.79 | 59.36 | 64.22 | 69.04 | 69.69 |
| | ManIQA↑ | 0.5269 | 0.6371 | 0.6018 | 0.5071 | 0.5338 | 0.6024 | 0.6215 |
| | ClipIQA↑ | 0.5609 | 0.7734 | 0.7004 | 0.6331 | 0.7263 | 0.6874 | 0.6843 |
| DRealSR | NIQE↓ | 5.3139 | 3.0885 | 4.1390 | 7.0159 | 5.5639 | 4.3661 | 4.1682 |
| | MUSIQ↑ | 34.68 | 36.18 | 34.51 | 30.52 | 32.79 | 37.22 | 40.30 |
| | ManIQA↑ | 0.4675 | 0.5985 | 0.5758 | 0.4210 | 0.4755 | 0.5797 | 0.5703 |
| | ClipIQA↑ | 0.5208 | 0.7568 | 0.6746 | 0.5884 | 0.7231 | 0.7540 | 0.6914 |

Table 1: Quantitative no-reference (NR) metrics comparison with state-of-the-art **DM-based** methods for Real-ISR (×4). The best and second-best results of each metric within both multi-step and one-step diffusion-based methods are highlighted in **red** and **blue**, respectively.

To intuitively demonstrate the effectiveness of EA-DISTS, we visualize the feature maps during the DISTS computation process. Figure 4 presents the visualization results of VGG-16 feature maps. As shown in Fig. 4, in areas rich with image details, such as the building windows, the feature maps associated with EA-DISTS exhibit more high-frequency information. Compared to DISTS, EA-DISTS demonstrates higher contrast in textured and smooth regions, further emphasizing the textural details within the images. Our EA-DISTS places greater emphasis on texture details within images, guiding the model to generate realistic and rich details.

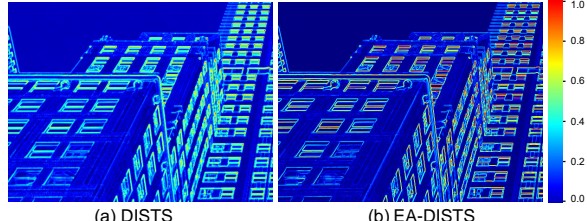

(a) DISTS  (b) EA-DISTS

Figure 4: Feature visualization associated with DISTS and EA-DISTS. Our EA-DISTS captures more high-frequency information, like texture and edges.

## 3.4 DISTILLATION-FREE TRAINING

Here, we summarize the whole distillation-free one-step diffusion model training process. As described in Section 3.1, within the generator component, DFOSD obtains $\hat{z}_H$ and the decoded high-resolution image $\hat{x}_H$ through one-step sampling. The generator then updates its parameters by computing the spatial loss $\mathcal{L}_{\text{spatial}}$ in pixel space between the generated image and the ground truth, as well as the adversarial loss $\mathcal{L}_\mathcal{G}$ derived from the discriminator in the latent space (Eq. 2). The loss function for updating the generator is defined as $\mathcal{L}_{\text{spatial}} + \lambda_1 \mathcal{L}_\mathcal{G}$. Specifically, we employ a weighted sum of Mean Squared Error (MSE) loss and perceptual loss to define the spatial loss:

$$\mathcal{L}_{\text{spatial}}(\mathcal{G}_\theta(x_L), x_H) = \mathcal{L}_{\text{MSE}}(\mathcal{G}_\theta(x_L), x_H) + \lambda_2 \mathcal{L}_{\text{EA-DISTS}}(\mathcal{G}_\theta(x_L), x_H), \quad (8)$$

where $\lambda_1$ and $\lambda_2$ are hyperparameters used to balance the contributions of each loss component.

For discriminator training, we utilize paired training features, where each pair consists of a negative sample feature $\hat{z}_H$ (generated by the generator) and the corresponding real image's latent representation $z_H$ as a positive one. Using Eq. 3, we compute the adversarial loss $\mathcal{L}_\mathcal{D}$ to update the discriminator's parameters. Furthermore, the discriminator can be initialized with weights from more powerful pre-trained models, such as SDXL (Podell et al., 2023), to achieve superior performance.

This distillation-free training approach allows our DFOSD to overcome the limitations imposed by multi-step diffusion models, enhancing generator performance without increasing its parameter count or compromising efficiency. Additionally, the integration of a robust discriminator initialized with advanced pre-trained models ensures that the generator receives high-quality feedback, facilitating the production of more realistic and detailed high-resolution images.

## 4 EXPERIMENTS

We conduct comprehensive experiments to validate the effectiveness of DFOSD in real-world image super-resolution (Real-ISR). We provide a detailed introduction of our experimental setup in Section 4.1. In Section 4.2, we evaluate our method on three challenging real-world datasets: RealSR (Cai et al., 2019), RealSet65 (Yue et al., 2024), and DRealSR (Wei et al., 2020), and compare it against the current state-of-the-art methods. In Section 4.3, We carry out comprehensive ablation studies to validate the effectiveness and robustness of our proposed approach.

| Datasets | Metrics | Non-Diffusion | | Multi-step Diffusion | | | | One-step Diffusion | | |
|---|---|---|---|---|---|---|---|---|---|---|
| | | Real-ESRGAN | SwinIR | StableSR-s50 | DiffBIR-s50 | SeeSR-s50 | ResShift-s15 | SinSR-s1 | OSEDiff-s1 | DFOSD-s1 |
| DRealSR | PSNR↑ | 30.55 | 28.31 | 30.31 | 25.91 | 28.35 | 26.42 | 27.33 | 24.20 | 26.47 |
| | SSIM↑ | 0.8571 | 0.8273 | 0.8394 | 0.6190 | 0.8052 | 0.7310 | 0.7237 | 0.7355 | 0.7838 |
| | LPIPS↓ | 0.3843 | 0.2736 | 0.2818 | 0.5347 | 0.3031 | 0.4582 | 0.4444 | 0.3429 | 0.3149 |
| | DISTS↓ | 0.2034 | 0.1387 | 0.1428 | 0.2387 | 0.1665 | 0.2382 | 0.2262 | 0.1763 | 0.1547 |
| RealSR | PSNR↑ | 27.57 | 27.34 | 26.28 | 24.87 | 26.20 | 25.45 | 25.83 | 24.57 | 24.60 |
| | SSIM↑ | 0.7741 | 0.7862 | 0.7733 | 0.6486 | 0.7555 | 0.7246 | 0.7183 | 0.7202 | 0.7221 |
| | LPIPS↓ | 0.2729 | 0.2515 | 0.2622 | 0.3834 | 0.2806 | 0.3727 | 0.3641 | 0.3036 | 0.3031 |
| | DISTS↓ | 0.1542 | 0.1583 | 0.2147 | 0.2015 | 0.1784 | 0.2344 | 0.2193 | 0.1808 | 0.1775 |

Table 2: Quantitative FR metrics comparison for Real-ISR (×4). The best and second-best results within both multi-step and one-step diffusion-based methods are highlighted in **red**, **blue**, respectively.

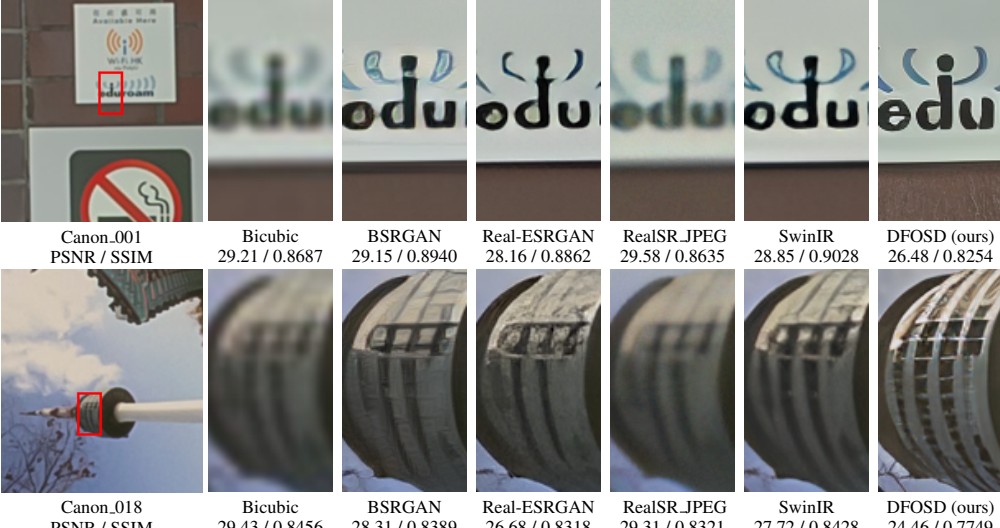

Canon_001 PSNR / SSIM    Bicubic 29.21 / 0.8687    BSRGAN 29.15 / 0.8940    Real-ESRGAN 28.16 / 0.8862    RealSR_JPEG 29.58 / 0.8635    SwinIR 28.85 / 0.9028    DFOSD (ours) 26.48 / 0.8254

Canon_018 PSNR / SSIM    Bicubic 29.43 / 0.8456    BSRGAN 28.31 / 0.8389    Real-ESRGAN 26.68 / 0.8318    RealSR_JPEG 29.31 / 0.8321    SwinIR 27.72 / 0.8428    DFOSD (ours) 24.46 / 0.7749

Figure 5: Visual comparison (×4) of DFOSD with GAN-based and Transformer-based methods. Cannon_001 contains the letters 'edu'. Cannon_018 contains the structures of tower windows. Although GAN-based approaches achieve higher PSNR and SSIM scores, their generated images exhibit less realistic and detailed textures compared to DFOSD. Those quantitative and visual comparisons indicate that higher PSNR and SSIM values do not mean better visual quality.

## 4.1 EXPERIMENTAL SETTINGS

**Datasets.** We train DFOSD on a self-collected dataset comprising 200K high-quality images. During training, we randomly crop patches of size 512×512 pixels from these images. To generate low-resolution (LR) and high-resolution (HR) pairs for training, we apply the Real-ESRGAN degradation pipeline. We conduct extensive evaluations of DFOSD on multiple real-world datasets, including RealSR (Cai et al., 2019), RealSet65 (Yue et al., 2024), and DRealSR (Wei et al., 2020). To avoid potential biases and ensure a fair comparison, we evaluate our model and all other methods by using the whole images from each dataset. We assess image quality without any cropping (*e.g.,* random crop, central crop) that might make the evaluation results randomly and hard to reproduce.

**Implementation Details.** We adopt Stable Diffusion (SD) 2.1-base as the backbone for training DFOSD, setting both the rank and scaling factor $\alpha$ of LoRA to 16 in the generator and discriminator. The model is trained using the AdamW optimizer with learning rates of $5\times10^{-5}$ for both generator and discriminator. We utilize a learnable text embedding as the conditional input for the SD UNet, without any prompts, and remove the text encoder. Training is performed with a batch size of 16 over 100K iterations with 4 NVIDIA A100-40GB GPUs.

**Compared Methods.** We compare our DFOSD with state-of-the-art diffusion model (DM)-based methods for real image super-resolution (Real-ISR), as well as other prominent approaches, including GAN-based and Transformer-based methods. The DM-based methods encompass multi-step diffusion models, such as StableSR (Wang et al., 2024a), ResShift (Yue et al., 2024), DiffBIR (Lin et al., 2024), and SeeSR (Wu et al., 2024b), alongside recently proposed one-step diffusion models like SinSR (Wang et al., 2024b) and OSEDiff (Wu et al., 2024a). OSEDiff is the current top-performing one-step diffusion Real-ISR method. Other methods include GAN-based approaches, such as BSRGAN (Zhang et al., 2021), RealSR-JPEG (Ji et al., 2020), and Real-ESRGAN (Wang et al., 2021b), as well as Transformer-based method SwinIR (Liang et al., 2021).

| | StableSR | DiffBIR | SeeSR | ResShift | SinSR | OSEDiff | DFOSD (ours) |
|---|---|---|---|---|---|---|---|
| # Step | 200 | 50 | 50 | 15 | 1 | 1 | 1 |
| Inference Time / s | 11.50 | 7.79 | 5.93 | 0.71 | 0.16 | 0.35 | **0.11** |
| # Total Param / M | $1.4 \times 10^3$ | $1.6 \times 10^3$ | $2.0 \times 10^3$ | 173.8 | 173.8 | $1.4 \times 10^3$ | 966.3 |
| # MACs / G | 75,812 | 24,528 | 32,336 | 4,903 | 2,059 | 2,269 | 2,132 |

Table 3: Complexity comparison ($\times 4$) among different methods, including sampling steps during inference, inference time, parameter count, and MACs. Inference time and MACs are tested for an output size of $512 \times 512$ with a single A100-40GB GPU.

| Dataset | NIQE↓ | MUSIQ↑ | ManIQA↑ | ClipIQA↑ |
|---|---|---|---|---|
| LSDIR + 10K FFHQ | 3.9264 | 67.26 | 0.6140 | 0.6397 |
| Our Dataset | **3.9255** | **69.21** | **0.6402** | **0.6683** |

Table 4: Quantitative comparison ($\times 4$) on RealSR. Our DFOSD is trained on different datasets.

**Evaluation Metrics.** To comprehensively assess the performance of each method, we employ four full-reference (FR) and four no-reference (NR) image quality metrics. The FR metrics consists of Peak Signal-to-Noise Ratio (PSNR), Structural Similarity Index Measure (SSIM), Learned Perceptual Image Patch Similarity (LPIPS) (Zhang et al., 2018a), and Deep Image Structure and Texture Similarity (DISTS) (Ding et al., 2020). PSNR measures pixel-wise differences, while SSIM evaluates structural similarity. Both PSNR and SSIM are computed on the Y channel in the YCbCr color space. LPIPS assesses perceptual similarity using deep neural network features. DISTS combines structural and textural comparisons. The NR metrics include Naturalness Image Quality Evaluator (NIQE) (Zhang et al., 2015), Multi-scale Image Quality Transformer (MUSIQ) (Ke et al., 2021), Multi-scale Attention-based Image Quality Assessment (ManIQA) (Yang et al., 2022), and ClipIQA (Wang et al., 2023a). NIQE evaluates image quality based on statistical features. MUSIQ captures multi-scale distortions using Transformers. ManIQA employs attention mechanisms to assess quality. ClipIQA leverages pre-trained models like CLIP to align quality assessments with human perception.

## 4.2 COMPARISON WITH STATE-OF-THE-ART METHODS

**Quantitative Results.** Tables 1 and 2 provide quantitative comparisons of the methods across the three datasets. DFOSD achieves either the best or second-best performance on the majority of metrics across all datasets when compared with other one-step diffusion methods. Although GAN-based methods outperform diffusion-based methods in terms of PSNR and SSIM, they generally exhibit poorer performance on NR metrics. Detailed comparisons of NR metrics and visual results for non-diffusion-based methods are provided in the **supplementary material**. Despite the higher FR metrics achieved by GAN-based and Transformer-based methods, their visual results are significantly inferior to those of DFOSD. Figure 5 illustrates several examples, further highlighting the limitations of full-reference metrics in accurately evaluating image quality. This underscores the necessity for more effective approaches to assess the quality of generated images.

**Visual Results.** Figure 6 presents a visual comparison of various diffusion-based Real-ISR methods. As observed, most existing methods struggle to generate realistic details and often produce incorrect content in certain regions of the image due to noise artifacts. Notably, our DFOSD demonstrates a significant advantage over others, particularly in the restoration of textual content. Additional visual comparison results are provided in the **supplementary material**.

**Complexity Analysis.** Table 3 presents a complexity comparison of DM-based Real-ISR methods, including the number of inference steps, inference time, parameter numbers, and MACs (Multiply-Accumulate Operations). All methods are evaluated on an NVIDIA A100 GPU. DFOSD achieves the fastest inference speed among all DM-based methods. Furthermore, since we do not employ a text encoder or other additional modules (such as DAPE used by OSEDiff and SeeSR, and ControlNet used by DiffBIR), our DFOSD has the smallest number of model parameters during inference among Stable Diffusion (SD)-based methods, reducing the parameters by 33% compared to OSEDiff.

## 4.3 ABLATION STUDY

**Training Data Scaling.** We train DFOSD on the LSDIR (Li et al., 2023) combined with the 10K FFHQ (Karras et al., 2024) dataset and our own collected high-quality dataset, respectively. We provide quantitative results in Table 4 and visual comparisons in Fig. 7. Our collected high-quality dataset provides rich priors, enhancing the authenticity and details.

Figure 6: Visual comparisons (×4) on Real-ISR task.

Figure 8: Visual results (×4) of DFOSD with different perceptual losses. The left side shows a comparison of the checkerboard. The right one shows content about some numbers, *i.e.*, '24, 26, 28'.

**Perceptual Loss.** Table 5 presents the impact of different perceptual loss functions, as well as the scenario where only Mean Squared Error (MSE) is applied as the spatial loss. Figure 8 showcases the visual outcomes of these experiments. The results indicate that incorporating perceptual loss is crucial for training SR models, as it facilitates the generation of more realistic details and enhances overall visual quality. Our proposed edge-aware DISTS (EA-DISTS) achieves the best performance across various image quality metrics and visual as-

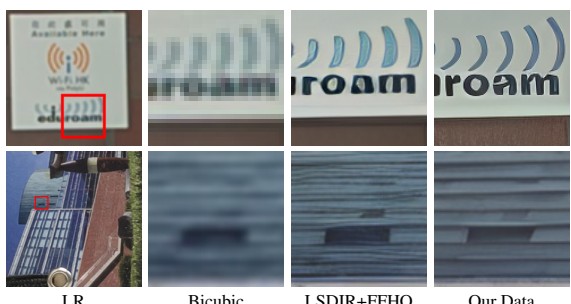

Figure 7: Visual comparison of DFOSD trained on LSDIR+10K FFHQ versus our high-quality dataset.

sessments. As shown in Fig. 8, EA-DISTS excels in producing highly realistic details, demonstrating its advantage in perceptual quality. This highlights the effectiveness of EA-DISTS in accurately restoring image textures and details, thereby significantly improving the visual quality.

| Loss Function | NIQE↓ | MUSIQ↑ | ManIQA↑ | ClipIQA↑ |
|---|---|---|---|---|
| MSE | 4.4463 | 65.35 | 0.5457 | 0.5833 |
| LPIPS | 4.3331 | 68.42 | 0.5914 | 0.6534 |
| EA-LPIPS | 4.1958 | 68.81 | 0.6077 | 0.6519 |
| DISTS | 4.2018 | 69.08 | 0.6223 | 0.6555 |
| EA-DISTS | **3.9255** | **69.21** | **0.6402** | **0.6683** |

Table 5: Impact of different perceptual loss functions on DFOSD performance.

| Discriminator | Base Model | NIQE↓ | MUSIQ↑ | ManIQA↑ | ClipIQA↑ |
|---|---|---|---|---|---|
| None | N/A | 6.9621 | 62.36 | 0.5597 | 0.5833 |
| Vanilla Discriminator | SD 2.1-base | 6.1392 | 64.36 | 0.5666 | 0.6059 |
| Diffusion-GAN Discriminator | SD 2.1-base | 4.5183 | 67.51 | 0.5800 | 0.6246 |
| NAD | SD 2.1-base | **3.9255** | 69.21 | 0.6402 | 0.6683 |
| NAD | SDXL 1.0-base | 4.0613 | **69.86** | **0.6870** | **0.6731** |

Table 6: Performance comparison of DFOSD with different discriminators. The best and second best results of each metric are highlighted in **red** and **blue**, respectively.

**Noise-Aware Discriminator (NAD).** We evaluate the impact of various discriminator modules on the training of DFOSD, including NAD, vanilla discriminator, diffusion-GAN (Wang et al., 2023b) style discriminator, and training without any discriminator. Both the vanilla and diffusion-GAN style discriminators are initialized with weights from the Stable Diffusion (SD) 2.1-base model (Rombach et al., 2022a), similar to the NAD described in Section 3.2. The experimental results, detailed in Table 6, indicate that the generator trained with NAD consistently outperform those utilizing other discriminators that are also initialized with SD 2.1-base. Specifically, NAD demonstrates superior capability in effectively guiding the generator, leading to improved image quality. This demonstrates the advantages of NAD in training distillation-free one-step diffusion models.

Additionally, we conduct experiments where the NAD is initialized with weights from the SDXL (Podell et al., 2023) model to further validate the effectiveness of our approach. As shown in the last two rows of Table 6, the NAD initialized with SDXL 1.0-base weights achieves superior performance compared to its counterparts, without requiring any modifications to the generator's architecture. This suggests that DFOSD can effectively leverage the strengths of more powerful pre-trained models, and enhance the performance of generator without compromising its efficiency.

# 5 DIFFERENCES WITH OTHER ONE-STEP DIFFUSION SR MODELS

We further discuss the difference between our DFOSD and representative one-step diffusion image SR methods, SinSR (Wang et al., 2024b) and OSEDiff (Wu et al., 2024a).

**Difference with SinSR. First**, SinSR requires performing multi-step deterministic sampling during training to obtain noise-image pairs, which greatly increases the training time. DFOSD does not rely on the results generated by the muti-step pre-trained diffusion models. **Second**, the involvement of a teacher model during training further escalates memory consumption. In contrast, each training iteration of DFOSD takes a lower latency than SinSR.

**Difference with OSEDiff. First**, OSEDiff leverages Variational Score Distillation (VSD) to optimize generated images, which necessitates the participation of 3 SD UNets during training, resulting in increased memory usage and prolonged training time. In comparison, our DFOSD requires only 1.5 SD UNets, reducing the training model size by at least $50\%$. **Second**, OSEDiff extracts prompts from LR images with DAPE, and encoding them into conditional input for SD UNet. DFOSD only uses learnable text embedding as the conditional input, which further reduce computational cost.

# 6 CONCLUSION

In this work, we propose DFOSD, a Distillation-Free One-Step Diffusion model, for Real-ISR. Departing from the diffusion distillation strategies commonly employed in previous studies, our approach effectively reduces training overhead. Specifically, we design a noise-aware discriminator (NAD) that capitalizes on the aggregation capabilities of intermediate features from a pre-trained SD UNet. NAD makes it hard for the generator to distinguish reconstruction from real images. Additionally, we propose the edge-aware DISTS (EA-DISTS) perceptual loss, which significantly enhances the texture realism and visual quality of the generated images. Our distillation-free strategy enables DFOSD to outperform pre-trained multi-step diffusion models in terms of visual results. Comprehensive experiments confirm that DFOSD achieves superior performance and substantially improves the realism of the generated images. These advancements highlight the potential of our method for more efficient and effective image restoration tasks.

## ETHICS STATEMENT

The research conducted in the paper conforms, in every respect, with the ICLR Code of Ethics.

## REPRODUCIBILITY STATEMENT

We have provided implementation details in Section 4.1. We will also release all the code and models.

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
