# ONE-STEP DIFFUSION-BASED REAL-WORLD IMAGE SUPER-RESOLUTION

## 1 COMPARISON WITH NON-DIFFUSION-MODEL-BASED NR IQA METHODS

Table 1 presents a quantitative comparison between DFOSD and several non-diffusion-model-based no-reference image quality assessment (NR IQA) methods, including BSRGAN, RealSR-JPEG, Real-ESRGAN, and SwinIR (Zhang et al., 2021; Ji et al., 2020; Wang et al., 2021; Liang et al., 2021). Across all evaluated datasets, DFOSD consistently outperforms these methods in NR IQA metrics. Specifically, while GAN-based methods like BSRGAN and Real-ESRGAN achieve competitive performance in traditional full-reference (FR) metrics such as PSNR and SSIM, they lag behind DFOSD in NR IQA metrics, which better capture perceptual quality aspects such as image clarity, quality, and detail.

| Datasets | Methods | NIQE↓ | MUSIQ↑ | MANIQA↑ | CLIPIQA↑ |
|---|---|---|---|---|---|
| DRealSR | BSRGAN | 4.6896 | 35.49 | 0.4650 | 0.5703 |
| | RealSR-JPEG | 7.4922 | 22.41 | 0.3183 | 0.4100 |
| | Real-ESRGAN | 4.7157 | 35.25 | 0.4767 | 0.5180 |
| | SwinIR | 4.6729 | 35.81 | 0.4617 | 0.5070 |
| | **DFOSD** | **4.1682** | **40.30** | **0.5703** | **0.6914** |
| RealSR | BSRGAN | 4.6609 | 63.59 | 0.5279 | 0.5436 |
| | RealSR-JPEG | 6.9524 | 36.07 | 0.3413 | 0.3612 |
| | Real-ESRGAN | 4.6917 | 59.68 | 0.5386 | 0.4899 |
| | SwinIR | 4.6864 | 59.63 | 0.5111 | 0.4652 |
| | **DFOSD** | **3.9255** | **69.21** | **0.6402** | **0.6683** |

Table 1: Performance comparison of DFOSD with non-diffusion-model-based NR IQA methods across three datasets. The best results for each metric among the methods are highlighted in **red**.

As shown in Table 1, DFOSD achieves superior performance across all NR IQA metrics compared to non-diffusion-model-based methods. Specifically, DFOSD exhibits lower NIQE scores, indicating higher perceptual quality and better image clarity. Additionally, DFOSD outperforms better in the MUSIQ, ManIQA, and ClipIQA metric, further demonstrating its ability to preserve and enhance image details. These results underscore the effectiveness of DFOSD in generating high-quality super-resolved images that maintain superior visual fidelity without relying on diffusion model-based architectures.

**Visual Comparison.** Figure 1 provides a visual comparison of images generated by DFOSD and the non-diffusion-model-based methods mentioned above. While GAN-based methods like BSRGAN and Real-ESRGAN produce visually appealing results, they often introduce artifacts and lack the fine-grained details that DFOSD preserves. In contrast, DFOSD consistently generates images with sharper edges, more accurate textures, and overall higher visual fidelity, aligning better with human perceptual judgments of image quality.

Despite the competitive performance of GAN-based and transformer-based methods in FR metrics, their NR IQA scores reveal shortcomings in capturing perceptual quality nuances. The superior performance of DFOSD in NR IQA metrics indicates its enhanced capability to generate images that are not only quantitatively superior but also qualitatively more pleasing to the human eye. This highlights the importance of incorporating NR IQA evaluations when assessing the true visual effectiveness of super-resolution models.

| $r$ | $\alpha$ | NIQE↓ | MUSIQ↑ | ManIQA↑ | ClipIQA↑ |
|-----|----------|-------|--------|---------|----------|
| 4 | 4 | 4.0909 | 68.50 | 0.6327 | 0.6521 |
| 8 | 8 | **3.9706** | **69.41** | **0.6365** | **0.6571** |
| 16 | 16 | **3.9255** | **69.21** | **0.6402** | **0.6683** |
| 8 | 64 | 9.4521 | 29.35 | 0.3298 | 0.2808 |
| 64 | 128 | 5.4717 | 65.42 | 0.5853 | 0.5517 |

Table 2: Impact of different LoRA rank and $\alpha$ on DFOSD performance.

| Conditional input | NIQE↓ | MUSIQ↑ | ManIQA↑ | ClipIQA↑ |
|-------------------|-------|--------|---------|----------|
| empty string | 4.2647 | 67.42 | 0.6291 | 0.6437 |
| DAPE extracted prompt | 4.0499 | **69.35** | **0.6453** | **0.6493** |
| random noise | **3.9899** | 69.17 | 0.6391 | 0.6373 |
| learnable text embedding | **3.9255** | **69.21** | **0.6402** | **0.6683** |

Table 3: Impact of different UNet conditional input on DFOSD performance.

## 2 ADDITIONAL ABLATION STUDIES

In this section, we present further ablation studies that complement those discussed in the main text.

**LoRA Settings.** We primarily investigate the impact of varying the $\alpha$ and rank settings of LoRA on the performance of DFOSD. The performance of DFOSD under different LoRA configurations is presented in Table 2. With lower $\alpha$ and rank values, the LoRA parameters are insufficient to achieve optimal results. As both $\alpha$ and rank increase, the fine-tuning capability of LoRA on the model is enhanced, leading to gradual improvements in performance. However, setting either $\alpha$ or rank too high results in significant overfitting, thereby degrading performance on the test set.

**UNet Conditional Input.** We investigate the impact of different conditional inputs for the UNet on the model's performance, including using an empty string as a prompt, employing DAPE to extract prompts from low-resolution (LR) images, utilizing random noise as a text embedding, and using a learnable text embedding. Table Table 3 presents the results of these experiments. Although DAPE shows significant advantages over using an empty string as a prompt, its performance is comparable to that of the learnable text embedding.

## 3 ALGORITHM OF DFOSD

The pseudo-code of our DFOSD training algorithm is summarized as Algorithm 1.

## 4 IMPLEMENTATION DETAILS

This section provides the implementation details of our **DFOSD**, including model hyperparameters, training procedures, and evaluation settings.

### 4.1 HYPERPARAMETER SETTINGS

During the training process, several key hyperparameters of DFOSD are crucial for achieving optimal performance. Table 4 summarizes these important hyperparameters used in our experiments.

### 4.2 EVALUATION DETAILS

We evaluate DFOSD and other methods on entire images from each test set. Following the implementations of StableSR and OSEDiff, we also apply the Adaptive Instance Normalization (AdaIN) algorithm to post-process generated images, ensuring that the color and style of the generated images closely match those of the input low-resolution (LR) images.

For evaluating large images, we adopt a tiling strategy to address memory limitations. Specifically, each image is divided into overlapping patches of size $512 \times 512$ pixels, with a 64-pixel overlap

---

**Algorithm 1** Training Algorithm for DFOSD

---

**Require:**
    $\epsilon_\phi$: Pretrained Stable Diffusion (SD) UNet
    $E_\phi$, $D_\phi$: Pretrained SD VAE Encoder and Decoder
    $\mathcal{S}$: Training dataset
    $N$: Number of training iterations
1: **Initialize** generator $\mathcal{G}_\theta$ from pretrained SD model:
    $E_\theta \leftarrow E_\phi$            ▷ Initialize encoder from SD VAE
    $\epsilon_\theta \leftarrow \epsilon_\phi$ with trainable LoRA      ▷ Initialize UNet with LoRA
    $D_\theta \leftarrow D_\phi$            ▷ Initialize decoder from SD VAE
2: **Initialize** guidance module $\mathcal{D}_\theta$ using downsampling and middle blocks from pretrained SD UNet
3: **for** $i = 1$ **to** $N$ **do**
4:      Sample a batch of $(x_L, x_H)$ from $\mathcal{S}$
    **/* Generator Step */**
5:      $z_L = E_\theta(x_L)$         ▷ Encode low-resolution image
6:      $\hat{z}_H = \frac{z_L - \sqrt{1 - \bar{\alpha}_{T_L}}\, \epsilon_\theta(z_L; T_L)}{\sqrt{\bar{\alpha}_{T_L}}}$         ▷ Denoising step
7:      $\hat{x}_H = D_\theta(\hat{z}_H)$         ▷ Decode high-resolution image
8:      $\mathcal{L}_{\text{spatial}} = L_{\text{MSE}}(x_H, \hat{x}_H) + \lambda_2 L_{\text{EA-DISTS}}(x_H, \hat{x}_H)$         ▷ Compute spatial loss
9:      Sample $t \in [0, T]$
10:      $\mathcal{L}_\mathcal{G} = -\mathbb{E}_{x_L \sim p_{\text{data}},\, t \sim [0,T]}\left[\log \mathcal{D}_\theta\left(F\left(\hat{z}_H, t\right)\right)\right]$         ▷ Compute generator adversarial loss
11:      Update $\mathcal{G}_\theta$ using $\mathcal{L}_{\text{spatial}} + \lambda_1 \mathcal{L}_\mathcal{G}$
    **/* Discriminator Step */**
12:      $z_H = E_\theta(x_H)$         ▷ Encode ground-truth high-resolution image
13:      Sample $t \in [0, T]$
14:      $\mathcal{L}_\mathcal{D} = -\mathbb{E}_{x_L \sim p_{\text{data}},\, t \sim [0,T]}\left[\log\left(1 - \mathcal{D}_\theta\left(F\left(\hat{z}_H, t\right)\right)\right)\right]$
15:      $\quad - \mathbb{E}_{x_H \sim p_{\text{data}},\, t \sim [0,T]}\left[\log \mathcal{D}_\theta\left(F\left(z_H, t\right)\right)\right]$
16:      Update $\mathcal{D}_\theta$ using $\mathcal{L}_\mathcal{D}$
17: **return** $\mathcal{G}_\theta$

---

| Hyperparameter | Value |
|---|---|
| Generator Learning Rate | $5 \times 10^{-5}$ |
| Discriminator Learning Rate | $5 \times 10^{-7}$ |
| Number of Training Iterations | 100,000 |
| Batch Size | 16 |
| Generator Adversarial Loss Weight ($\lambda_1$) | $5 \times 10^{-3}$ |
| EA-DISTS Loss Weight ($\lambda_2$) | 1 |

Table 4: Key hyperparameters for training DFOSD.

between adjacent patches to ensure smooth transitions. We perform inference independently on each image patch and subsequently stitch them together. For the overlapping regions, we average the results to maintain consistency and continuity across the entire image.

The models used for each evaluation metric are listed in Table 5. All metrics are computed using the `pyiqa` library. For PSNR and SSIM, we evaluate the Y channel in the YCbCr color space of the images to focus on luminance information, which is more indicative of perceived image quality.

## 5 LIMITATIONS AND DISCUSSIONS

While our method demonstrates promising results, it has certain limitations. **Firstly**, we have not yet incorporated recently proposed larger diffusion models, such as SDXL, as the base models for the generator. Consequently, the effectiveness of our approach on large-scale models remains to be validated. **Secondly**, our method employs a fixed guidance scale during training. Although the guidance scale is less critical for super-resolution tasks, users may still desire the flexibility to adjust the generation intensity in specific scenarios.

| Metric | Model File |
|--------|-----------|
| LPIPS | `LPIPS_v0.1_alex-df73285e.pth` |
| DISTS | `DISTS_weights-f5e65c96.pth` |
| MUSIQ | `musiq_koniq_ckpt-e95806b9.pth` |
| ManIQA | `MANIQA_PIPAL-ae6d356b.pth` |
| ClipIQA | `RN50.pt` (CLIP module) |

Table 5: Models used for each evaluation metric. All metrics are computed using the `pyiqa` library. For PSNR and SSIM, evaluations are performed on the Y channel in the YCbCr color space.

## 6 MORE VISUAL COMPARISONS

Figures 2, 3 presents additional visual comparison results with compared methods (Wang et al., 2024a; Yue et al., 2024; Lin et al., 2024; Wu et al., 2024b; Wang et al., 2024b; Wu et al., 2024a; Zhang et al., 2021; Ji et al., 2020; Wang et al., 2021; Liang et al., 2021). Our DFOSD demonstrates superior visual quality, detail, and realism in highly degraded scenarios, fine hair details, text, and richly textured regions.

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

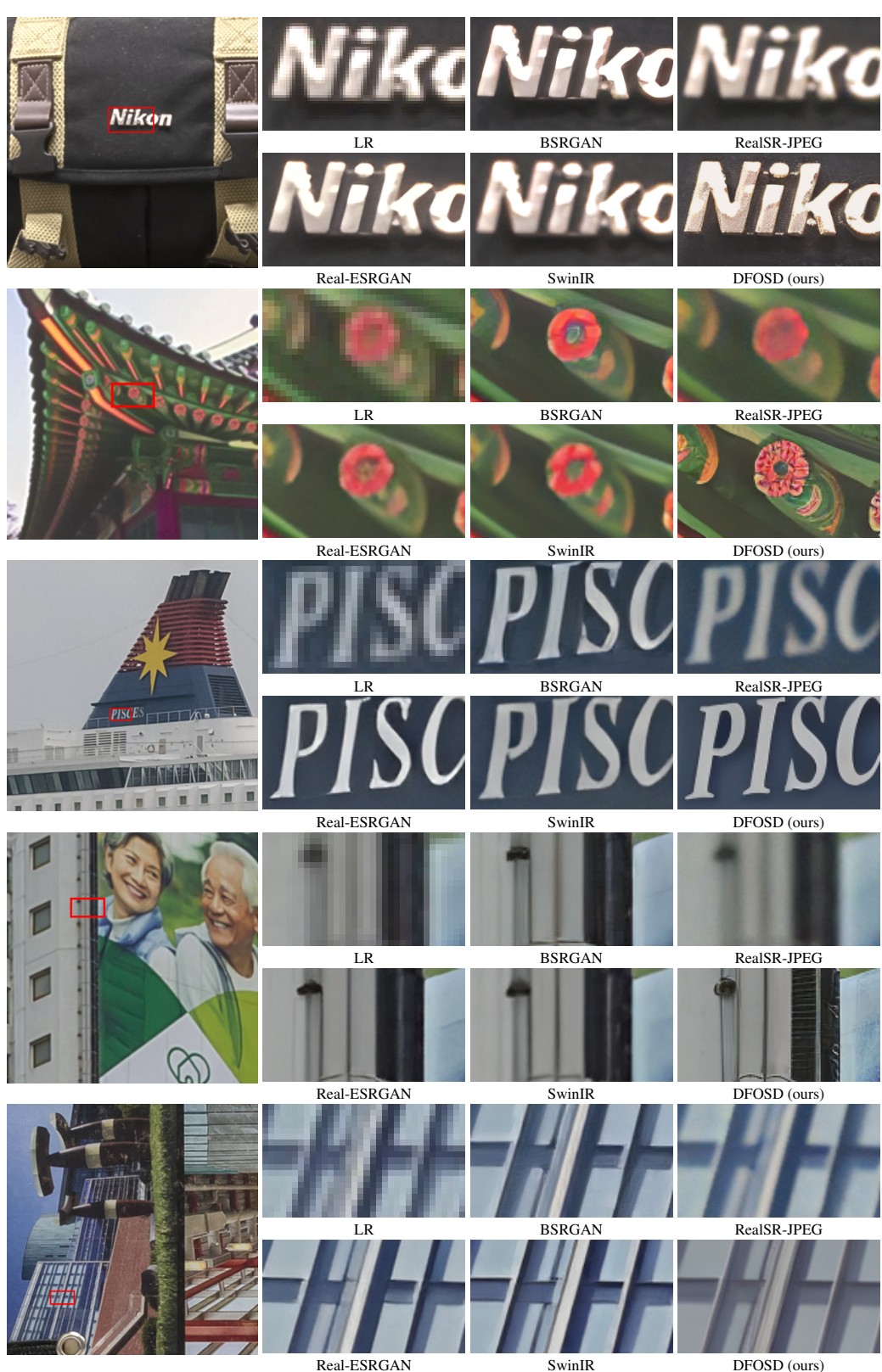

Figure 1: Visual comparison of super-resolved images generated by DFOSD and non-diffusion-model-based methods. DFOSD produces images with sharper edges and more realistic textures, demonstrating superior perceptual quality.

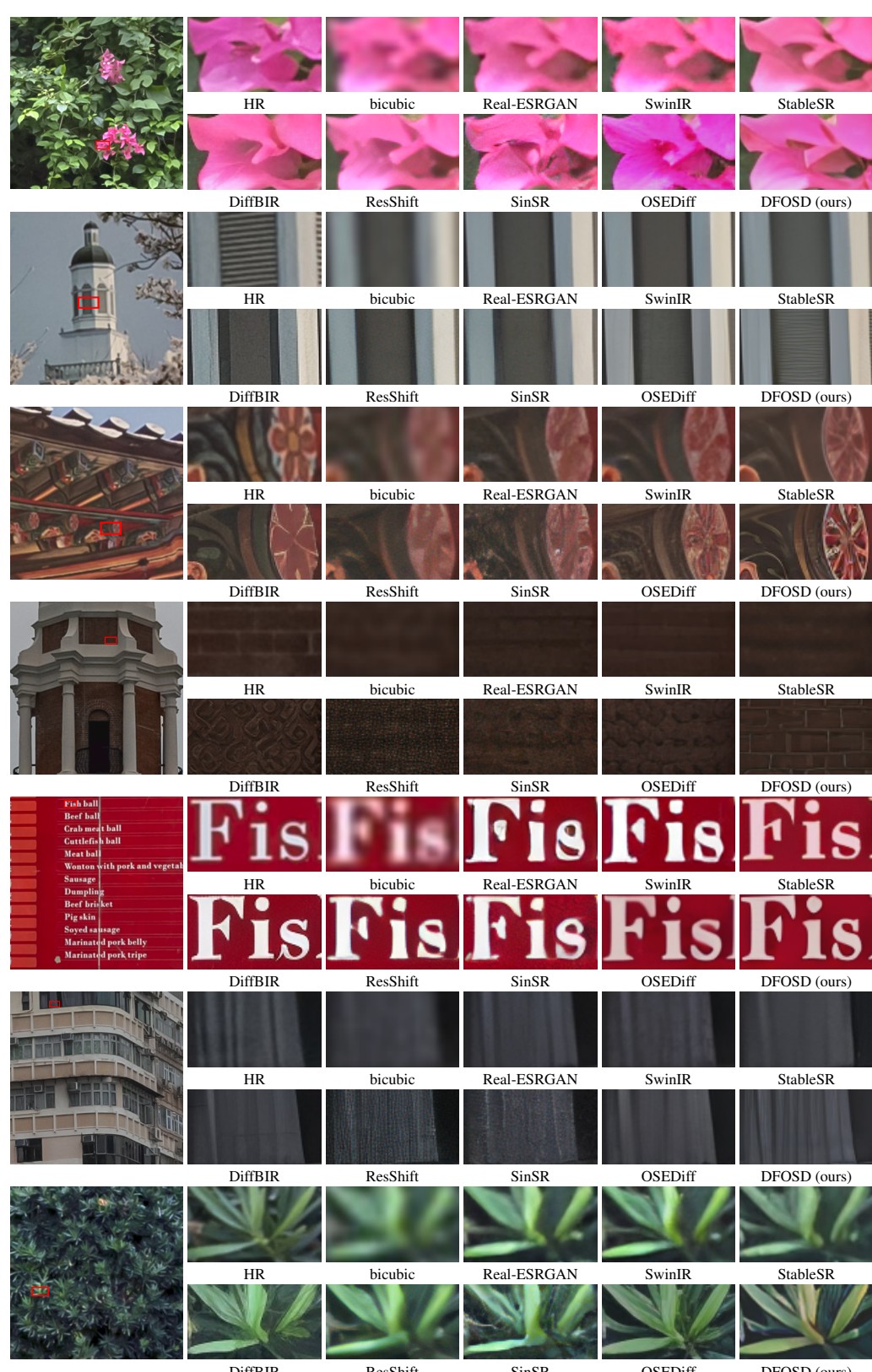

Figure 2: More visulization comparisons of different DM-based Real-ISR methods. Zoom in for best view.

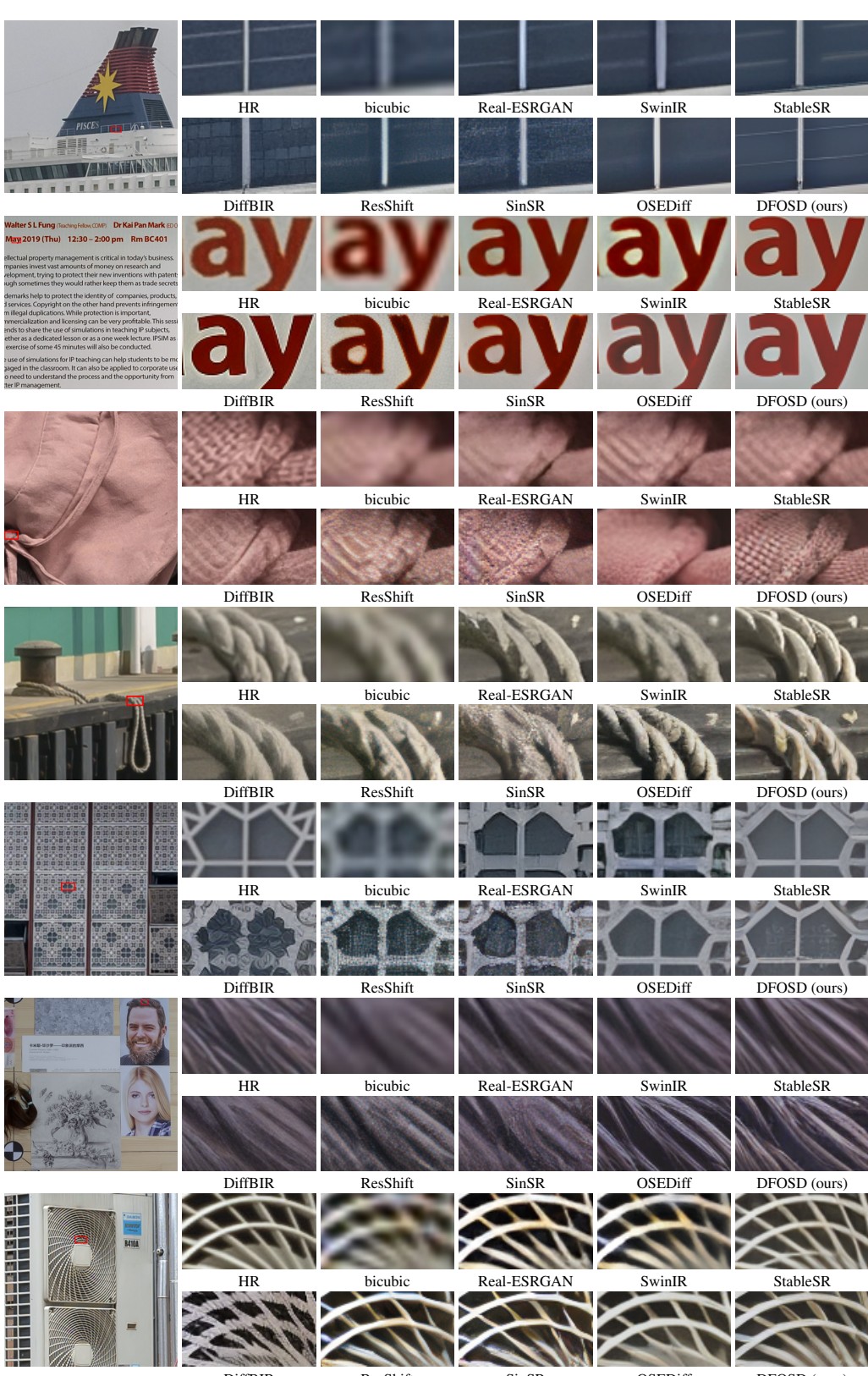

Figure 3: More visulization comparisons of different DM-based Real-ISR methods. Zoom in for best view.