# OpenReview forum: "Distillation-Free One-Step Diffusion for Real-World Image Super-Resolution"
_ICLR.cc/2025/Conference — ICLR 2025 Conference Withdrawn Submission_

### Official Review · Reviewer_ULbS · 2024-10-31

**Soundness:** 2
**Presentation:** 3
**Contribution:** 2
**Rating:** 5
**Confidence:** 4

**Summary:**

This paper introduces DFOSD, a one-step diffusion model for real-world image super-resolution that bypasses multi-step diffusion processes and teacher models, reducing training and inference time. It integrates a noise-aware discriminator (NAD) within an adversarial framework to boost perceptual SR quality and employs an EA-DISTS loss to further enhance perceptual performance.

**Strengths:**

1. Quantitative and qualitative analyses clearly demonstrate the effectiveness of the proposed method. Specifically, Figure 3 illustrates how DFOSD successfully aligns mid-level features with real image distributions. DFOSD achieves significant improvements in both distortion-based metrics (PSNR and SSIM) and perceptual metrics, which is interesting. Additionally, computational costs are significantly reduced, as shown in Table 3.

**Weaknesses:**

1. The relationship between the proposed NAD and EA-DISTS remains somewhat unclear. Both components aim to enhance perceptual performance, but it would be beneficial for the reviewer if their complementary relationship, if any, were explicitly clarified.

2. Although Table 5 provides ablation studies on different loss functions, other perceptual losses should be included for a more comprehensive comparison. The table currently highlights the superiority of DISTS over LPIPS, but this might be due to the larger parameters used in DISTS. It would be useful to include additional perceptual losses, such as NIQE, MUSIQ, ManiQA, and ClipIQA, in both their original and EA-enhanced versions.

3. What distinguishes NAD from *? What specific advantages does NAD offer over these approaches?

*A. Sauer, Adversarial diffusion distillation
*A. Sauer, Fast High-Resolution Image Synthesis with Latent Adversarial Diffusion Distillation

4. Since this paper follows the Real-ESRGAN degradation pipeline, it can use any high-quality images for training, as shown in Table 4. However, as this is not a unique contribution of the paper, it would be helpful, if any, to include detailed information on "our dataset."

**Questions:**

1. While NAD operates on noisy latents domain, an alternative approach would involve operating on decoded images. The reviewer acknowledges that the VAE decoder has a large parameter count, yet it would be insightful to see experimental results in the image domain.

2. As Weakness 4, could the authors provide details about the collected dataset, specifically regarding its scale, resolution, and diversity?

---

### Official Review · Reviewer_BymW · 2024-10-31

**Soundness:** 3
**Presentation:** 3
**Contribution:** 3
**Rating:** 5
**Confidence:** 5

**Summary:**

This paper proposes DFOSD, a Distillation-Free One-Step Diffusion SR model that enhances image detail and visual quality. Key contributions include a Noise-Aware Discriminator (NAD), which improves realism through adversarial training, and Edge-Aware DISTS (EA-DISTS) loss, which leverages image edges to enhance the authenticity of reconstructed details.

**Strengths:**

1. This paper proposes a noise-aware discriminator, leveraging the prior knowledge from the pre-trained SD UNet. This enhances the realism and details of the reconstructed images without much memory usage and training time.

2. The proposed EA-DISTS can enhance texture detail restoration.

3. The writing is well, and the idea is easy to follow.

**Weaknesses:**

1. This paper introduces learnable text embedding to replace the text extractor, significantly reducing inference time. How it is implemented and trained and more explanation of learnable text embedding are needed for clarity.

2. This paper evaluates image quality without cropping (Sec. 4.1, Lines 362-364), which is unusual for comparing SD-based SR methods, as they are sensitive to input resolution. I suggest evaluating the methods on the pre-cropped test dataset from StableSR [1] (https://huggingface.co/datasets/Iceclear/StableSR-TestSets), which has a fixed resolution with $512\times512$ avoiding random crop and non-reproducible results. This test dataset is widely used in various SD-based SR methods, ensuring a more standardized and fair comparison while addressing the authors' concerns.

[1] Wang, Jianyi, et al. "Exploiting diffusion prior for real-world image super-resolution." International Journal of Computer Vision (2024): 1-21.

3. The idea of NAD is similar to UFOGen [2] and LADD [3]. Relevant references and comparisons should be provided.

[2] Xu, Yanwu, et al. "Ufogen: You forward once large scale text-to-image generation via diffusion gans." Proceedings of the IEEE/CVF Conference on Computer Vision and Pattern Recognition. 2024.

[3] Sauer, Axel, et al. "Fast high-resolution image synthesis with latent adversarial diffusion distillation." arXiv preprint arXiv:2403.12015 (2024).

**Questions:**

1. DFOSD uses learnable text embedding without DAPE and the text encoder, reducing inference time compared to OSEDiff. However, it's unclear if this fully accounts for the 0.24s speedup. The authors should provide a breakdown of inference times for each major component (e.g., text embedding, main network, etc.) for DFOSD and OSEDiff on the same device. This would help clarify where the speedup is coming from.

2. In Table 4, DFOSD's performance with the LSDIR+10K FFHQ training dataset is worse than OSEDiff with the same training dataset in no-reference metrics (MUSIQ, ManIQA, ClipIQA). It would be useful to clarify if these improvements in no-reference metrics are primarily due to the high-quality training dataset. A more detailed analysis in Sec. 4.3 would be helpful.
To avoid the influence of input resolution, I suggest the authors evaluate the DFOSD's performance with different training datasets on the pre-cropped test dataset (https://huggingface.co/datasets/Iceclear/StableSR-TestSets) from StableSR [1].

[1] Wang, Jianyi, et al. "Exploiting diffusion prior for real-world image super-resolution." International Journal of Computer Vision (2024): 1-21.

I will consider raising my score if my primary concerns are addressed.

---

### Official Review · Reviewer_ge9q · 2024-11-01

**Soundness:** 2
**Presentation:** 2
**Contribution:** 2
**Rating:** 3
**Confidence:** 5

**Summary:**

This paper introduces a model named DFOSD, addressing the problem of real-world image super-resolution. The authors propose a noise-aware discriminator (NAD) and an edge-aware DISTS (EA-DISTS) loss to optimize the model, resulting in superior performance on quantitative metrics and qualitative assessments. DFOSD achieves remarkable results on tasks such as image restoration, demonstrating significant improvements in realism and detail generation across various real-world datasets.

**Strengths:**

+ The paper presents DFOSD, a novel model that significantly advances real-world image super-resolution by offering a distillation-free one-step diffusion approach, which is highly innovative in the field.

+ Two standout contributions are the noise-aware discriminator (NAD) and the edge-aware DISTS (EA-DISTS) loss. The NAD leverages prior knowledge from pre-trained models to enhance realism, while EA-DISTS improves texture detail restoration.

+ The writing is clear and methodical and the experimental section is robust, providing not only quantitative metrics but also qualitative assessments that demonstrate DFOSD's superior performance and efficiency in image super-resolution tasks.

**Weaknesses:**

- Usually, the training cost is not very large for diffusion-based SR methods compared to text-2-image tasks, so I think the distillation-free optimization is not much necessary.  Besides, we also could pre-compute the output of teacher models with multi-steps predictions before starting the complete training.  Can you elaborate further on the advantages of non-distillation training?

- The DFOSD proposed in this paper is just a marginal optimization based on OSEDiff[1] and other adversail training-based methods[2,3,4].

- The proposal of EA-DISTS loss lacks of novelty, just an experimental trick.

- Noise-aware discriminator is not new,  the same ideas are shown in SD3-turbo[2] and TAD-SR[3]. Although the NAD seems simpler and effective, but is not a very innovative method.

- The experimental setting is not rigorous and unfair,  will you release the 200K high-quality images to public?


[1] One-Step Effective Diffusion Network for Real-World Image Super-Resolution, 2024.

[2] Adversarial Diffusion Distillation, 2023.

[3] Fast High-Resolution Image Synthesis with Latent Adversarial Diffusion Distillation, 2024.

[4] One Step Diffusion-based Super-Resolution with Time-Aware Distillation, 2024.

**Questions:**

Please referring to the weaknesses above.

---

### Official Review · Reviewer_k26t · 2024-11-01

**Soundness:** 3
**Presentation:** 3
**Contribution:** 2
**Rating:** 6
**Confidence:** 4

**Summary:**

This paper presents a new GAN-based real-world image super-resolution method (Real-ISR) using pretrained diffusion models. The work introducesa Noise-Aware Discriminator (NAD) and an edge-aware perceptual loss function (EA-DISTS) for the GAN training. The paper presents extensive experimental results demonstrating that the proposed method achieves superior performance in both quantitative metrics and visual quality compared to state-of-the-art diffusion-based and GAN-based methods for Real-ISR.

**Strengths:**

1. The idea of using the pretrained diffusion model to train a real-sr GAN is new. The introduction of the Noise-Aware Discriminator (NAD) and the edge-aware DISTS (EA-DISTS) perceptual loss seems novel and effective.
2. Comprehensive experimental results on three real-world datasets show the proposed DFOSD achieves competitive or superior performance in both no-reference (NR) and full-reference (FR) image quality metrics.
3. The overall writing is good and the paper is easy to understand.

**Weaknesses:**

1. Although the authors claim that the proposed method, DFOSD (Distillation-Free One-Step Diffusion), is a diffusion SR model, it is essentially a GAN-based method. The model only uses the parameters trained by a diffusion model, but there is no Markov process. This term may cause some misunderstanding of the method.
2. While the paper emphasizes the reduction in training overhead and computational complexity relative to distillation-based methods, the overall framework still relies on heavy pre-trained models (e.g., Stable Diffusion UNet). The method may not be as lightweight as simpler GAN-based approaches, which could limit its adoption in resource-constrained environments. A more explicit comparison with simpler non-diffusion-based methods in terms of memory and computational requirements would provide a clearer picture.
3. Although the authors report visual comparisons and use several no-reference and full-reference metrics, the paper would benefit from subjective user studies to evaluate the perceived quality of the generated high-resolution images.
4. The paper does not provide an analysis of how sensitive DFOSD is to hyperparameter choices, such as the weights of the loss function components.

**Questions:**

How does this method compare to traditional GAN ​​methods (Real-ESRGAN, BSRGAN) in terms of running costs?

---

### Note · Authors · 2024-11-14

I have read and agree with the venue's withdrawal policy on behalf of myself and my co-authors.